# Predictability of Radiologically Measured Psoas Muscle Area for Intraoperative Hypotension in Older Adult Patients Undergoing Femur Fracture Surgery

**DOI:** 10.3390/jcm12041691

**Published:** 2023-02-20

**Authors:** Youn Young Lee, Jae Hee Woo, In-Young Yoon, Hyun Jung Lee, Sang-Mee Ahn, Ji Seon Chae, Youn Jin Kim

**Affiliations:** 1Department of Anesthesiology and Pain Medicine, Ewha Womans University Seoul Hospital, Seoul 07804, Republic of Korea; 2Department of Anesthesiology and Pain Medicine, College of Medicine, Ewha Womans University, Seoul 07804, Republic of Korea; 3Department of Anesthesiology and Pain Medicine, Ewha Womans University Mokdong Hospital, Seoul 07985, Republic of Korea

**Keywords:** frailty, sarcopenia, hip fracture, psoas muscle, older adult patients, hypotension

## Abstract

This retrospective study aimed to determine the predictive value of radiologically measured psoas muscle area (PMA) for intraoperative hypotension (IOH) using receiver operating characteristic (ROC) curves in older adult patients with hip fractures. The cross-sectional axial area of the psoas muscle was measured by CT at the level of the 4th lumbar vertebrae and normalized by body surface area (BSA). The modified frailty index (mFI) was used to assess frailty. IOH was defined as an absolute threshold of mean arterial blood pressure (MAP) < 65 mmHg or a relative decrease in MAP > 30% from baseline MAP. Among the 403 patients, 286 (71.7%) had developed IOH. PMA normalized by BSA in male patients was 6.90 ± 0.73 in the no-IOH group and 4.95 ± 1.20 in the IOH group (*p* < 0.001). PMA normalized by BSA in female patients was 5.18 ± 0.81 in the no-IOH group and 3.78 ± 0.75 in the IOH group (*p* < 0.001). The ROC curves showed that the area under the curve for PMA normalized by BSA and modified frailty index (mFI) were 0.94 for male patients, 0.91 for female patients, and 0.81 for mFI (*p* < 0.001). In multivariate logistic regression, low PMA normalized by BSA, high baseline systolic blood pressure, and old age were significant independent predictors of IOH (adjusted odds ratio: 3.86, 1.03, and 1.06, respectively). PMA measured by computed tomography showed an excellent predictive value for IOH. Low PMA was associated with developing IOH in older adult patients with hip fractures.

## 1. Introduction

Hip fractures are one of the most important causes of disability in the aging population and lead to an increasing socioeconomic health burden [1]. Surgical outcomes of patients with hip fractures are associated with a high 3-month mortality rate (4.7–19.5 %) and physical morbidity [2,3,4]. Therefore, identifying the predisposing risk factors preoperatively is important to improve surgical outcomes. Preoperative assessment of frailty and sarcopenia, as a surrogate of frailty, have emerged as useful predictors of surgical mortality and morbidity in various surgical conditions in older adults [5,6,7,8].

Frailty is defined as a state of vulnerability to poor resolution of homeostasis, characterized by unintentional weight loss, self-reported exhaustion, muscle weakness, slow walking speed, and low physical activity [9]. Sarcopenia, a component of frailty, is defined as the progressive loss of muscle mass and muscle strength [10]. Several frailty assessment tools have been suggested to predict surgical outcomes [11,12,13]; however, there is no consensus on the measurement of frailty. Assessing frailty using frailty assessment tools in patients with femur fractures is not always feasible because most have limited physical performance, and some are difficult to communicate with, such as those with neurocognitive or hearing disorders.

Recently, several studies have measured the cross-sectional area of the psoas muscle using computed tomography (CT) imaging as a validated method for quantifying sarcopenia [14,15,16]. Since pelvic bone CT was performed as a standard diagnostic work-up before surgery, the psoas muscle area (PMA) could be a useful candidate for preoperative risk evaluation along with frailty assessment tools.

Intraoperative hypotension (IOH) is associated with mortality and adverse postoperative outcomes, such as acute kidney injury [17], myocardial injury [18], and stroke [19]. IOH is associated with postoperative outcomes and frequently occurs in frail patients; they typically have a higher sympathetic drive and reduced baroreflex sensitivity, which leads to IOH [20,21]. The association between low PMA and adverse surgical outcomes in older adult patients with hip fractures has been described [5]. To the best of our knowledge, no study has demonstrated the association between sarcopenia, presented as low PMA, and IOH development. Therefore, we hypothesized that CT-measured PMA could predict IOH in older adult patients undergoing hip fracture surgery.

## 2. Materials and Methods

### 2.1. Study Design and Patients

This retrospective study was approved by the Institutional Review Board of Ewha Woman’s University Hospital (IRB no. 2022-04-039), and the requirement for written informed consent was waived. The data were collected from electronic medical records (EMR) of older adult patients (aged > 65 years) who underwent hip fracture surgery, such as arthroplasty and osteosynthesis, from January 2020 to December 2021 at two hospitals of Ewha Woman’s University (Seoul and Mokdong Hospital). The surgical procedure for hip fractures was performed by an orthopedic surgeon at each institution in the same medical school. Patients who underwent surgery under spinal and combined spinal epidural anesthesia, multiple fracture surgery, or had previous hip surgery in whom the cross-sectional area of the psoas could not be obtained using CT images because of artifacts from devices (e.g., metallic hip or spine prostheses), and those with incomplete or follow-up loss data were excluded. Of the 649 patients, 246 were excluded based on these criteria, and 403 were included (Figure 1).

### 2.2. Measurement of Psoas Muscle Area

The cross-sectional axial area (cm^2^) of the bilateral psoas muscle at the level of the fourth lumbar vertebra (L4) was assessed from the axial slice image of pelvic bone CT using picture archiving and communication systems (PACS) (Maroview 5.4, Infinitt, Seoul, Republic of Korea), as described previously [5,22]. Pelvic bone CT was performed as part of the patient’s routine diagnostic work-up, as ordered by the surgeon at our institution. Bilateral PMAs that were outlined manually at the L4 pedicle level by the freehand region of interest (ROI) program in PACS (Figure 2) were averaged and divided by the body surface area (BSA) for normalization according to Canales et al. [14]. The PMAs from the axial CT image were measured by two anesthesiologists retrospectively: one (Y.Y.L.) measured all images, and the other who was blinded to the outcomes (I.Y.) measured randomly selected 50 images. The inter-class correlation coefficient (ICC) was used to measure inter-observer agreement. The PMA value normalized by BSA was divided into five groups. The highest quintile (≥5.48) was considered as a reference to examine factors associated with developing IOH and 3-month unfavorable outcomes using logistic regression analysis.

### 2.3. Assessment of Frailty and Intraoperative Hypotension

All patients were assessed for frailty using the modified frailty index (mFI) based on EMRs. The mFI originated from the frailty index of the Canadian Study of Health and Aging Frailty Index [11] by matching 70 variables to 11 categories of comorbidities and deficits. Factors of mFI were obtained from the patient’s medical history, which consisted of 11 deficit variables (Appendix A) calculated by summation of variables and divided by 11. We categorized the patients into two groups: the non-frail (including pre-frail) (0 ≤ mFI < 0.27) and frail groups (mFI ≥ 0.27) based on previous studies [23]. In the logistic regression analysis for predicting IOH and 3-month unfavorable outcomes, we used mFI categorical groups by dividing 11 groups based on the mFI score (0 to 1).

The baseline blood pressure was defined as the first value of non-invasive blood pressure (systolic, diastolic, and mean blood pressure) after entering the operating room in the supine position. Radial artery cannulation was conducted under sedation before or right after induction, which was electronically recorded every five minutes (up to 1 min intervals in case of describing in detail) using BESTcare 2.0 (ezCaretech, Seoul, Republic of Korea) program. In some cases in which attending anesthesiologists wanted to describe the hypotensive blood pressure in detail, it was recorded up to 1 min intervals. If the patients showed 30% higher baseline blood pressure than the average blood pressure during the hospitalization period, we allowed the blood pressure to be lowered after premedication with sedatives according to our institutions’ routine. In this case, blood pressure before the induction was considered baseline blood pressure. We defined IOH as the absolute threshold of mean arterial blood pressure (MAP) < 65 mmHg or a relative decrease in MAP > 30% from the baseline, which lasted for at least 1 min [18,24]. For regression analysis, the incidence of IOH was analyzed dichotomously. Data on managing IOH, such as the presence of vasopressors (ephedrine, phenylephrine bolus, or norepinephrine infusion), were also collected from anesthetic charts. Perioperative management was conducted using a standardized hip fracture anesthesia protocol in two hospitals in the same branch of medical school. In all patients, general anesthesia was induced using propofol (1–1.5 mg/kg) and fentanyl (1 µg/kg) and was maintained by inhalation agents (sevoflurane or desflurane). Our standardized protocol included maintenance of normovolemia during anesthesia and management of systolic blood pressure within 20% of the baseline by fluid and/or vasopressors [25]. The transfusion target was hemoglobin 8 g/dL for most of the patients, excluding the patients who needed higher targets (hemoglobin ≥ 10 g/dL) for transfusion [26].

### 2.4. Data Collection

Demographics, medical history, and perioperative laboratory findings were obtained from the EMRs at two institutions: age, sex, body mass index (BMI), American Society of Anesthesiologists (ASA) classification, and comorbidities. It also included variables such as diabetes mellitus (controlled by diet alone or treated with oral antihyperglycemic therapy or insulin), hypertension (HTN) on medication, type of hypertensive medication, cerebrovascular disease (CVA), transient ischemic accident (TIA), heart diseases (arrhythmia, congestive heart failure, ischemic heart disease, previous myocardial infarction within 6 months, or coronary intervention/bypass graft at any time), pulmonary diseases (pneumonia or chronic obstructive pulmonary disease exacerbation within 30 days), peripheral vascular disease, and cognitive impairment, which were used to calculate mFI. The following preoperative laboratory findings were collected: hemoglobin, platelets, white blood cells, serum albumin, C-reactive protein, blood urea nitrogen, and serum creatinine. Intraoperative variables were collected from the EMRs, including baseline and lowest blood pressure (non-invasive and invasive), presence of IOH, operation type, anesthesia time, operation time, input (intraoperative crystalloid and colloid infusions or blood transfusion) estimated blood loss, and presence of vasopressors. Postoperative data were collected: intensive care unit (ICU) admission, length of hospital stay, presence of postoperative delirium, postoperative deep vein thrombosis, postoperative pneumonia, and CVA during hospitalization. Rehospitalization (including admission to a nursing hospital), death, or falls within 3 months after surgery were regarded as 3-month adverse outcomes using logistic regression analysis [27].

### 2.5. Study Outcomes

The primary outcome was the predictive value of PMA normalized by BSA for IOH during hip fracture surgery in older adult patients using receiver operating characteristic (ROC) curves. The secondary outcomes were the predictability of IOH by frailty using the mFI score and the association between PMA and frailty measured by the mFI score. Moreover, we analyzed the predictors of developing IOH and the 3-month unfavorable outcomes in older adult patients undergoing hip fracture surgery.

### 2.6. Statistical Analysis

Continuous variables were analyzed using an independent *t*-test or Mann–Whitney U test after assessment for normality using the Shapiro–Wilk test and are presented as mean ± standard deviation or median (interquartile range), as appropriate, whereas categorical variables were analyzed using χ^2^ tests or Fisher’s exact tests (if > 20% of the expected frequencies were < 5) and were presented as percentages. The ICC with a 95% confidence interval (CI) was calculated by two researchers to determine the reliability of the PMA measurements. ROC curve analysis was used to identify the predictability of PMA normalized by BSA in male and female patients (primary outcome), and frailty measured by the mFI score was used to discriminate with or without IOH groups (secondary outcome). The data were presented as the area under the curve (AUC) with 95% CI. We suggested cut-offs of PMA normalized by BSA for IOH and frailty by sex using Youden’s index and by respective sensitivity and specificity. The Hosmer–Lemeshow goodness-of-fit test and AUC were used to assess the model fit. The association between frailty and PMA normalized by BSA, the factors associated with developing IOH, and 3-month unfavorable outcomes were analyzed using a binary logistic regression model (secondary outcome). Confounding factors, including age, sex, albumin level, and ASA classification (<III or ≥III), were identified from previous studies [28]. Additionally, significant variables, including 11 variables in mFI, were assessed by comparing patients with and without IOH. We also conducted inversed probability of treatment weighting (IPTW) to adjust for confounding factors [29]. Covariates which showed a significant difference in Table 1 were used for calculating propensity scores. Multivariate logistic regression was used to analyze the predictive factors for IOH and 3-month unfavorable outcomes using the backward selection method. In this process, sex and age were considered, and major factors were selected after checking for multicollinearity. The data were presented as area odds ratios (ORs) with 95% CI. Statistical analyses were performed using Microsoft Excel 2010 (Microsoft Corp., Redmond, WA, USA), MedCalc Statistical Software Version 20.1.4 2020 (MedCalc Software Bvba, Ostend), and International Business Machine Statistical Package for the Social Sciences Statistics (version 22.0; IBM Corp., Armonk, NY, USA). Statistical significance was assumed at *p* < 0.05, and two-tailed *p*-values were used.

## 3. Results

As presented in Table 1, 403 patients were analyzed, of which 286 (71.7%) had developed IOH. The patients had a mean age of 81.34 ± 8.60 years, and 64% (*n* = 258) of the patients had ASA ≥ III. The population underwent hip surgery, including open reduction internal fixation, hip (*n* = 186; 46.2%), bipolar hemiarthroplasty (*n* = 123; 30.5%), and total hip replacement (*n* = 76; 18.8%). The emergency surgery portion in the IOH group was 20.3%, which was higher than that of 12% in the no-IOH group (*p* = 0.048). The number of patients with chronic HTN was higher in the IOH group than that in the non-IOH group (231 [73.1%] vs. 85 [26.9%]; *p* < 0.001). Most patients were on medication, and the largest percentage of the combined drugs were angiotensin-converting enzyme inhibitors (angiotensin receptor blockers) and β-blockers. The baseline systolic blood pressure was higher in the IOH group (153.45 ± 20.61 vs. 141.46 ± 20.13; *p* < 0.001). Patients in the IOH group were supplemented with a higher intravenous volume to manage hypotension (*p* = 0.035). Preoperative laboratory findings were not significantly different between the groups.

Frailty and postoperative outcomes are presented in Table 2. PMA normalized by BSA in male patients was 6.90 ± 0.73 in the no-IOH group and 4.95 ± 1.20 in the IOH group (*p* < 0.001). PMA normalized by BSA in female patients was 5.18 ± 0.81 in the no-IOH group and 3.78 ± 0.75 in the IOH group (*p* < 0.001). The ICC between the two researchers for measuring PMA was excellent: 0.977 (95% CI, 0.95–0.99; *p* < 0.001), and measuring PMA from CT images took only less than 1 min. mFI score was significantly higher in the IOH group (0.34 ± 0.15) than that in the no-IOH group (0.17 ± 0.14) (*p* < 0.001). Frail patients were more included in the IOH group (*n* = 263; 92.3%) than in the no-IOH group (*n* = 14; 12%) (*p* < 0.001). The association between the five categorical groups of PMA normalized by BSA and frailty measured by mFI score was analyzed using logistic regression; OR was 2.735 (95% CI, 2.20–3.40; *p* < 0.001).

Patients in the IOH group showed a higher rate of ICU admission (35%; *n* = 100) than those in the no-IOH group (14.5%; *n* = 17; *p* < 0.001). After being weighted by estimated propensity scores, 125 patients (33.2%) in IOH group and 57 patients (14.5%) in no-IOH group showed a rate of ICU admission (*p* < 0.001). Longer duration of hospital stay was higher in the IOH group (15.02 ± 10.00 days) than that in the no-IOH group (11.96 ± 6.08 days; *p* = 0.006). After being weighted by estimated propensity scores, the duration of hospital stay was significantly longer in the IOH group (14.60 ± 11.63 vs. 12.46 ± 6.36 days; *p* = 0.002). The rates of 3-month adverse outcomes were significantly higher by 35.3% (*n* = 101) in the IOH group than that in the no-IOH group by 10.2% (*n* = 12) (*p* < 0.001). After being weighted by estimated propensity scores, the rates of 3-month adverse outcomes were 37.1% (*n* = 140) in the IOH group and 15.6% (*n* = 62) in the no-IOH group (*p* < 0.001).

The ROC curves (Figure 3) show that the AUC for PMA normalized by BSA and mFI were 0.94 for male patients (95% CI, 0.87–0.96; *p* < 0.001), 0.91 for female patients (95% CI, 0.87–0.96; *p* < 0.001), and 0.81 for mFI (95% CI, 0.77–0.85; *p* < 0.001). The optimal cut-off values of PMA normalized by BSA for predicting IOH were 6.17 (87.8% sensitivity and 92.9% specificity) and 4.50 (89.0% sensitivity and 88.1% specificity) for male and female patients, respectively.

The factors associated with IOH are presented in Table 3. Univariate analysis showed that old age, low albumin, hemoglobin levels, high baseline systolic blood pressure, low PMA normalized by BSA, and a greater number of mFI variables were significantly associated with IOH. The highest OR was found in patients with a low PMA normalized by BSA (OR 3.85; 95% CI, 2.92–5.07; *p* < 0.001), followed by the number of mFI variables (OR 2.07; 95% CI, 1.71–2.52; *p* < 0.001). Among the mFI variables, a history of CHF, HTN, TIA, CVA with sequelae, pulmonary diseases, cardiac intervention, and patients’ dependent functional status were significantly associated with IOH. Multivariate logistic regression revealed that low PMA (OR 3.86; 95% CI, 2.82–5.29; *p* < 0.001) normalized by BSA, high baseline systolic blood pressure (OR 1.03; 95% CI, 1.01–1.05; *p* = 0.001), and old age (OR 1.06; 95% CI, 1.02–1.10; *p* = 0.003) were significant independent predictors of IOH, whereas the number of mFI variables was not a significant independent predictor of IOH (*p* = 0.870).

As presented in Table 4, female sex, ASA classification (<III vs. ≥III), low albumin and hemoglobin levels, presence of IOH, low PMA normalized by BSA, a greater number of categorical mFI variables, and history of TIA were significantly shown to be associated with 3-month adverse outcomes using univariate logistic regression analysis. The results of multivariate logistic regression showed the highest OR for low PMA normalized by BSA (OR 1.62; 95% CI, 1.37–1.91; *p* < 0.001). In addition, old age (OR 1.08; 95% CI, 1.04–1.12; *p* < 0.001) and low albumin levels (OR 0.44; 95% CI, 0.23–0.85; *p* = 0.014) were significantly associated with 3-month unfavorable outcomes, whereas presence of IOH was not a significant independent predictor of 3-month unfavorable outcomes after adjusting for confounders (*p* = 0.478).

## 4. Discussion

In this retrospective study, 71.7% of patients experienced hypotensive events during femur fracture surgery. The patients in the IOH group had a lower PMA normalized by BSA than those in the no-IOH group, both in male and female patients. The present study demonstrated that preoperative PMA measured by CT showed an excellent AUC value for predicting IOH in older adult patients undergoing femur fracture surgery. Our results suggest that PMA obtained from pre-existing CT could be used as a simple and feasible method for predicting IOH in older adult patients with hip fractures who are unable to access frailty. It was also significantly associated with 3-month unfavorable outcomes.

Several previous studies have shown that radiologically assessed PMA, as a surrogate of sarcopenia, was related to surgical mortality in various surgical settings, such as transcatheter aortic valve implantation [6], abdominal [7], and femur fracture surgery [5,22]. Based on these results, we considered the underlying relationship between surgical mortality and low PMA. Frail patients have reduced baroreflex sensitivity and vagal function, which may predispose them to IOH [21]. Soysal et al. [30] showed the relationship between the severity of sarcopenia and orthostatic hypotension accompanied by decreased function of their cardiovascular function to maintain blood pressure. Hence, in this context, we hypothesized that CT-measured PMA is not only associated with adverse surgical outcomes but could also be related to IOH in older adult patients undergoing hip fracture surgery.

PMA normalization is necessary to consider its clinical use because it differs according to race, sex, and age. Several PMA normalization methods have been used, such as the psoas: L4 vertebrae index [5,6], which is normalized by BMI, BSA [12], and height [7]. Some studies using CT-measured PMA used height^2^ for adjustment; the present study utilized BSA, according to Canales et al. [14]. They demonstrated that the AUC value of PMA (cm^2^) normalized by BSA (m^2^) was 0.95 (95% CI, 0.89–1.00) for discriminating frailty measured by Fried phenotype frailty assessments, which is higher than that of the BMI 0.80 (95% CI, 0.65–0.95). Similarly, our study showed excellent AUC values using PMA normalized by BSA to discriminate IOH preoperatively in male (0.94 (95% CI, 0.87–0.96; *p* < 0.001)) and female patients (0.91 (95% CI, 0.87–0.96; *p* < 0.001)).

The American College of Surgeons and the American Geriatrics Society recommend routine preoperative evaluation of frailty for all patients aged > 65 years [31]. Beggs et al. [32] reported that adverse surgical outcomes increased two to three-fold in frail patients, regardless of the type of surgery and frailty assessment tools. However, it is difficult to use preoperative frailty assessment clinically because its questionnaire-based method is limited in its application for patients with femur fractures who have restricted physical conditions or inability to provide a medical history owing to altered mental status. Furthermore, no consensus tools or time-consuming measuring procedures have contributed to the restrictions on its routine use. PMA can be readily obtained from axial images of pelvic bone CT preoperatively, and it could be a feasible tool for patients with hip fractures to optimize perioperative risk assessment.

In the present study, we found that PSA normalized by BSA was superior to mFI score in predicting IOH. Measuring PMA from CT images before surgery could be a faster and more simple method than mFI for assessing the risks of IOH. Predicting IOH could improve individual anesthetic management, in terms of preparing for hypotensive situations: meticulous titration of anesthetic agents, preparation of additional peripheral intravenous lines, vasopressors, sufficient pre-induction fluid supplement, and continuous arterial blood pressure monitoring to prevent hypotensive events during the surgery. As a surrogate of frailty before surgery, measuring PMA from CT images took < 1 min and showed a relatively excellent ICC, suggesting it to be an individual prognostic tool for developing IOH.

Predicting IOH is difficult because there are distinct entities of hypotension depending on the phase of anesthesia, including vasodilation by induction agents, massive bleeding, greater anesthesia depth, patient’s hypovolemic state, and preoperative use of antihypertensive medications (angiotensin-II receptor blockers and angiotensin-converting enzymes). From our retrospective data, we found that low PMA, advanced old age, and high baseline systolic blood pressure were associated with the development of IOH, of which most IOH events occurred at three-time points: post-induction period, position-changing period supine to lateral or vice-versa, and before-emergence period. Consistent with our study, Sudfeld et al. [28] defined post-induction period hypotension in which the time within 20 min after induction was most prevalent and critical for anesthesiologists since surgical factors can be ruled out at this point. They suggested factors associated with post-induction and early IOH, including emergency surgery, low pre-induction systolic arterial pressure, and advanced age. Consistent with this study, we found that older patients were likely to have more IOH episodes and that more frail patients were included in the IOH group than that in the no-IOH group. Old age is associated with frailty, which leads to IOH. In contrast, emergency surgery, low pre-induction systolic arterial pressure, ASA class IV, and male sex were not associated with IOH development in our study population. This discrepancy may be attributed to the different populations of the study and the definition of IOH. Regarding the population with hip fractures, older women were more prone to falls than older men [33]. The patients had higher baseline blood pressure owing to pain caused when they were transferred to the operating bed. Moreover, Sudfeld et al. considered IOH only within the post-induction period and early 30 min after the start of surgery, and the incidence of hypotension was approximately 34%, which is approximately half of that in our study. This difference may be attributed to the use of a narrow definition of IOH and the absolute threshold of systolic blood pressure (<90 mmHg) for considering IOH in 30 years younger study population. Interestingly, ASA classification and the number of mFI variables did not show significant associations with IOH prediction and 3-month adverse outcomes in multivariate logistic regression analysis. We assumed that these tools have limitations because they are primarily used for the accumulation of underlying diseases. Radiologically measured PMA showed a closer relationship to functional physical status than ASA classification and mFI variables. Considering the impact of sarcopenia, PMA is related to active daily life [34], and it dynamically reflects patients’ physical status. This result indicates that sarcopenia and frailty are interrelated conditions that lead to IOH and adverse outcomes.

Several possible mechanisms may explain the development of IOH in frail patients. First, frailty is associated with impaired autonomic cardiovascular regulation. Decreased sympathetic vasoconstrictor nerve activity in frail patients induces profound vasodilatory effects by propofol induction agents [35] and could develop IOH, particularly during the post-induction period. Reduced baroreceptor responsiveness, represented by decreased heart rate variability, has been previously demonstrated in frail patients [36]. An impaired autonomic nervous system may affect the ability to maintain systemic blood pressure, which is important for cardiovascular homeostasis under stress. Second, the decreased water content in patients with low PMA could lead to hypotension during surgery. The water content in skeletal muscle is approximately 10% to 20% [37], and skeletal muscle mass, as a reservoir of water content, is important for maintaining blood pressure [38]. Low PMA, a surrogate of a sarcopenia-induced low water reservoir, may contribute to impaired blood pressure homeostasis in frail patients. Third, patients with pre-existing HTN were more vulnerable to IOH. Multivariate analysis showed that the adjusted OR of baseline systolic blood pressure for predicting IOH was 1.03, and the number of patients with chronic hypertension was higher in the IOH group than in the no-IOH group. Jor et al. [39] also demonstrated that the degree of preoperative hypertensive systolic blood pressure is associated with the development of IOH. Older patients with HTN are likely to be labile regarding systemic blood pressure. Moreover, the patients continued renin-angiotensin-aldosterone system antagonists till the day before surgery, which may be associated with a higher risk of IOH.

This study had several limitations. First, this was a retrospective study, and unfound confounding factors may exist. We conducted this study from two institutions of the same medical school using standardized anesthesia protocols and excluded neuraxial anesthesia to minimize selection bias due to the different pathophysiology of hypotension, which is attributed to depression of the sympathetic trunk. Despite these efforts, it has inherent limitations according to its design. Second, we could not measure the time-weighted average IOH from restricted data that automatically recorded arterial blood pressure from EMR at 1- to 5-min intervals. As longer exposure to IOH may be related to an increased risk of postoperative outcomes, future prospective studies should be done to clarify the association between IOH and 3-month adverse outcomes considering the time-weighted average IOH. Third, we analyzed the older adult femur-fractured cohort whose distribution was not equal to that of male and female patients because older women have lower bone density and are more prone to falls than older men [33]. Caution is necessary for generalizing our results, which should be validated in a multicenter prospective study in the future. Fourth, we only used the mFI score to assess frailty, which consisted of comorbidities or deficits that could be retrospectively assessed, unlike questionnaire-based assessment methods [10,11]. However, it is difficult to apply questionnaire-based frailty assessment tools to patients with femur fractures because of their physical disability. In this context, we attempted to suggest a feasible surrogate for frailty in patients with femur fractures, such as PMA, from existing pelvic bone CT. Finally, ICC was not measured for the entire cohort, and PMA was measured independently by anesthesiologists in randomly selected patients; PMA was measured by a trained anesthesiologist, not a radiologist. Nonetheless, the measurement of PMA was easy to learn, and the ICC was known to be high: 0.968 (95% CI, 0.961–0.973) for the right and 0.915 for the left (95% CI, 0.898–0.929) [22]. This advantage could be a strength of PMA for easy application in clinical settings by anesthesiologists.

## 5. Conclusions

PMA normalized by BSA using pre-existing CT was a significant predictor of IOH occurrence and unfavorable surgical outcomes in older adult patients with hip fractures. Moreover, preoperative simple measurement of PMA was superior to mFI in predicting IOH. To discuss the relationship between IOH and adverse surgical outcomes, further prospective studies using the time-weighted average IOH are warranted.

## Figures and Tables

**Figure 1 jcm-12-01691-f001:**
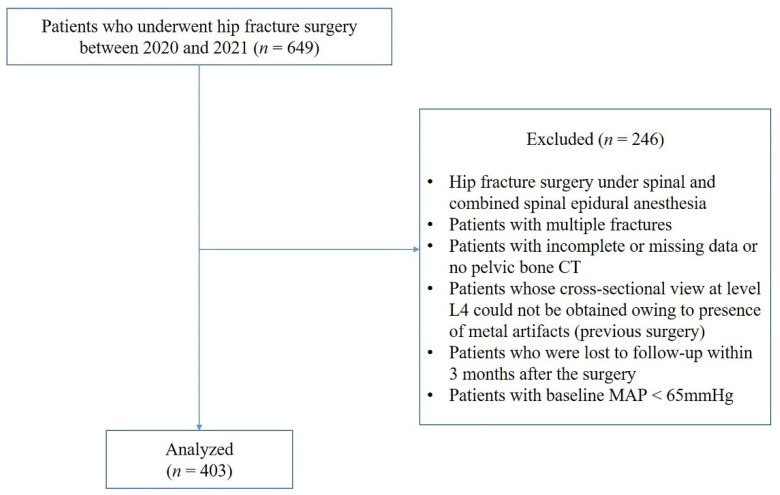
Study flow chart.

**Figure 2 jcm-12-01691-f002:**
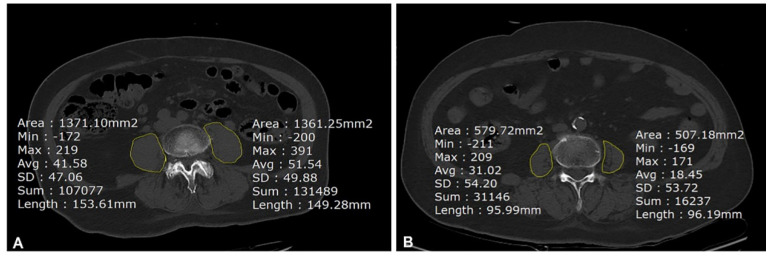
Cross-sectional images of the psoas muscle in pelvic bone CT. Cross-sectional image from the pedicle level of L4 axial view of pelvic bone CT. Right and left psoas muscle areas are measured using free ROI and averaged. Cross-sectional image of psoas muscle for (**A**) measured PMA of the patients in the 5th quintile group and (**B**) measured PMA of the patients in the 1st quintile group. CT, computed tomography; ROI, region of interest; PMA, psoas muscle area; BSA, body surface area.

**Figure 3 jcm-12-01691-f003:**
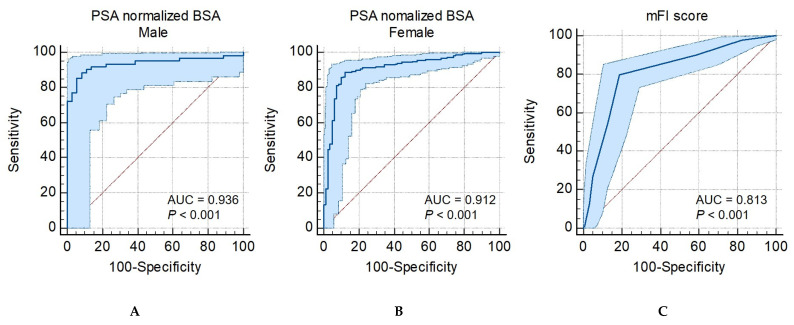
Receiver-operating characteristic curves for discriminating intraoperative hypotension ROC curve for predicting IOH using (**A**) PSA normalized by BSA in male patients, (**B**) PSA normalized by BSA in female patients, and (**C**) mFI score. ROC; receiver operating characteristic AUC; area under the curve; IOH, intraoperative hypotension; PMA, psoas muscle area; BSA, body surface area; mFI, modified frailty index.

**Table 1 jcm-12-01691-t001:** Demographic and baseline characteristics of participants.

Demographics	All Patients (*n* = 403)	No-IOH (*n* = 117)	IOH (*n* = 286)	*p*-Value
Age, year	81.34 ± 8.60	78.62 ± 8.43	82.46 ± 8.45	**0.001**
Sex, female	305 (75.7)	81 (69.1)	224 (78.3)	**0.053**
BMI, kg/m^2^	22.15 ± 3.76	21.62 ± 2.89	21.77 ± 3.35	0.997
ASA status				0.372
<III	145 (36)	46 (39.3)	99 (34.6)	
≥III	258 (64)	71 (60.7)	187 (65.4)	
Chronic arterial hypertension	316 (78.4)	85 (26.9)	231 (73.1)	**<0.001**
No medication	5 (1.6)	1 (1.2)	4 (1.6)	
ARB	41 (12.9)	9 (10.6)	32 (13.9)	
Ca^+2^ antagonist	127 (39.8)	35 (41.2)	92 (39.8)	
β-Blocker	16 (5.0)	2 (2.3)	14 (6.1)	
ACE inhibitor	2 (0.6)	0 (0)	2 (0.9)	
Diuretics	6 (1.9)	1 (1.2)	5 (2.2)	
Combined drugs	117 (38.2)	37 (43.5)	82 (35.5)	
DM	142 (35.2)	40 (34.2)	102 (35.6)	0.196
Surgical category				0.726
Open reduction internal fixation	186 (46.2)	54 (46.2)	132 (46.2)	
Closed reduction internal fixation	18 (4.5)	4 (3.4)	14 (4.9)	
Bipolar hemiarthroplasty	123 (30.5)	32 (27.4)	91 (31.8)	
Total hip replacement	76 (18.8)	27 (23)	49 (17.1)	
Emergency surgery	72 (17.9)	14 (12)	58 (20.3)	**0.048**
Preoperative data				
Hemoglobin, g/dL	11.53 ± 7.08	11.41 ± 1.90	10.92 ± 1.95	0.113
Platelet, 10^9^/L	209.84 ± 80.76	212.35 ± 82.06	208.74 ± 80.34	0.719
WBC, 10^3^/uL	8.90 ±3.21	8.91 ±2.97	8.90 ±3.32	0.992
Albumin, g/dL	3.67 ± 0.28	3.67 ± 0.28	3.67 ± 0.28	0.243
CRP, mg/dL	2.50 ± 3.00	2.50 ± 3.00	2.50 ± 3.00	0.361
BUN, mg/dL	22.29 ± 7.09	22.29 ± 7.09	22.29 ± 7.09	0.596
Creatinine, mg/dL	0.85 ±0.51	0.86 ±0.48	0.84 ±0.52	0.705
Intraoperative data				
Baseline SBP	149.86 ± 21.16	141.46 ± 20.13	153.45 ± 20.61	**<0.001**
Baseline DBP	82.16 ± 14.78	79.85 ± 12.71	83.15 ± 15.51	0.068
Baseline MBP	104.40 ± 15.15	102.39 ± 12.79	106.11 ± 15.77	0.052
Lowest SBP	92.20 ± 13.90	99.18 ± 14.38	89.21 ± 12.59	**<0.001**
Lowest DBP	50.96 ± 10.15	57.12 ± 10.51	48.33 ± 8.79	**<0.001**
Lowest MBP	64.50 ± 10.32	71.14 ± 10.10	61.68 ± 9.05	**<0.001**
Input, mL, (IQR)	801.01 (762.40, 839.62)	792.66 (748.56, 836.76)	820.22 (741.90, 898.55)	**0.035**
Estimated blood loss, mL, (IQR)	227.83 (209.69, 245.96)	243.17 (204.07, 282.26)	221.16 (201.28, 241.04)	0.903
Anesthesia time, min	114.03 ± 29.61	107.24 ± 26.41	114.03 ± 27.30	0.612
Operation time, min	60.21 ± 24.81	57.44 ± 22.17	58.94 ± 22.81	0.649

Values are expressed as means ± standard deviations, medians (interquartile ranges, IQR), or absolute numbers (percentages). BMI, body mass index; ASA, American Society of Anesthesiologists; DM, diabetes mellitus; HTN, hypertension; ARB, angiotensin receptor blocker; ACE, angiotensin-converting enzyme; DM, diabetes mellitus; WBC, white blood cell; CRP, C-reactive protein; BUN, blood urine nitrogen; Cr, creatinine; SBP, systolic blood pressure; DBP, diastolic blood pressure; MBP, mean blood pressure. *p*-value < 0.05 are in bold.

**Table 2 jcm-12-01691-t002:** Patients’ frailty assessment and postoperative 3-month outcomes before and after IPTW.

	All Patients (*n* = 403)			No-IOH (*n* = 117)	IOH (*n* = 286)		*p*-Value
PMA							
Male	9.35 ± 2.75			11.55 ±3.01	8.20 ±2.48		**<0.001**
Female	6.20 ± 1.64			7.81 ±1.65	5.68 ±1.63		**<0.001**
PMA normalized by BSA							
Male	5.63 ± 1.48			6.90 ± 0.73	4.95 ± 1.20		**<0.001**
Female	4.13 ±1.03			5.18 ± 0.81	3.78 ± 0.75		**<0.001**
mFI score	0.29 ± 0.17			0.17 ± 0.14	0.34 ±0.15		**<0.001**
Not-frail (mFI < 0.27)	125 (31)			103 (88)	23 (7.7)		**<0.001**
Frail (mFI ≥ 0.27)	278 (69)			14 (12)	263 (92.3)		**<0.001**
	**Before IPTW adjustment**			***p*-value**	**After IPTW adjustment**		***p*-Value**
	**All patients** **(*n* = 403)**	**No-IOH** **(*n* = 117)**	**IOH** **(*n* = 286)**		**No-IOH** **(*n* = 394)**	**IOH** **(*n* = 377)**	
ICU admission	117 (29)	17 (14.5)	100 (35)	**<0.001**	57 (14.5)	125 (33.2)	**<0.001**
Hospital length of stay, days	14.13 ± 10.13	11.96 ± 6.08	15.02 ±10.00	**0.006**	12.46 ± 6.36	14.60 ± 11.63	**0.002**
Delirium	138 (34.3)	34 (29.3)	104 (36.4)	0.177			
DVT	119 (29.9)	31 (27.2)	88 (31)	0.455			
pneumonia	50 (12.4)	10 (20)	40 (14)	0.140			
CVA	14 (3.5)	1 (0.9)	13 (4.5)	0.068			
3-month outcomes	113 (28.0)	12 (10.2)	101 (35.3)	**<0.001**	62 (15.6)	140 (37.1)	**<0.001**
Rehospitalization	86 (21.3)	10 (25.6)	76 (27.2)	**0.001**	51 (13)	109 (28.9)	**<0.001**
Death	10 (2.5)	0	10 (3.5)	**0.046**	4 (1.0)	12 (3.3)	**0.030**
Falls	17 (4.2)	2 (1.7)	15 (5.4)	0.099			

Values are expressed as means ± standard deviations, medians, or absolute numbers (percentages). IPTW, inverse probability if treatment weighting; mFI, modified frailty index; PMA, psoas muscle area; BSA, body surface area; ICU, intensive care unit, DVT, deep vein thrombosis; CVA, cerebrovascular disease. *p*-value <0.05 are in bold.

**Table 3 jcm-12-01691-t003:** Unadjusted and adjusted odds ratios for predicting IOH by binary logistic regression.

Variables	Unadjusted Analysis		Adjusted Analysis	
	Odds Ratio (95% CI)	*p*-Value	Odds Ratio (95% CI)	*p*-Value
Age (for 1 year increase)	1.047 (1.02, 1.08)	**0.002**	1.06 (1.02, 1.10)	0.003
Sex (female)	1.45 (0.84, 2.49)	0.182	1.23 (0.58, 2.60)	0.580
ASA <III vs. ≥III	1.25 (0.79, 1.90)	0.372	0.68 (0.33, 1.42)	0.305
Emergency	1.02 (0.55, 1.88)	0.945	—	—
Albumin	0.49 (0.30, 0.82)	**0.007**	0.76 (0.40, 1.45)	0.400
Hemoglobin	0.83 (0.72, 0.94)	**0.004**	1.08 (0.87, 1.33)	0.512
Baseline SBP	1.03 (1.02, 1.04)	**<0.001**	1.03 (1.01, 1.05)	**0.001**
PMA normalized by BSA (Reference: 5th quintile)	3.85 (2.92, 5.07)	**<0.001**	3.86 (2.82, 5.29)	**<0.001**
mFI variables (categorical)	2.07 (1.71, 2.52)	**<0.001**	1.01 (0.85, 1.21)	0.870
DM	2.39 (1.39, 4.13)	0.502	—	—
CHF	2.97 (1.55, 5.68)	**0.001**	—	—
HTN	3.00 (1.75, 5.13)	**<0.001**	—	—
TIA or CVA	4.25 (2.27, 7.95)	**<0.001**	—	—
Dependent functional status	8.67 (4.99, 15.04)	**<0.001**	—	—
MI	3.61 (0.81, 16.03)	0.091	—	—
Peripheral disease	2.23 (0.89, 5.56)	0.086	—	—
CVA with sequelae	2.79 (1.26, 6.18)	**0.011**	—	—
Pulmonary disease	3.58 (1.63, 7.86)	**0.001**	—	—
Cardiac intervention or angina	3.60 (1.37, 9.47)	**0.009**	—	—
Impaired sensory	1.13 (0.52, 2.44)	0.761	—	—

CI, confidence interval; ASA, American Society of Anesthesiologists; SBP, systolic blood pressure; mFI, modified frailty index; PMA, psoas muscle area; BSA, body surface area; mFI, modified frailty index; DM, diabetes mellitus; CHF, congestive heart failure; HTN, hypertension; TIA, transient ischemic attack; CVA, cerebrovascular disease; MI, myocardial infarction.

**Table 4 jcm-12-01691-t004:** Unadjusted and adjusted odds ratios for predicting 3-month unfavorable outcomes by binary logistic regression.

Variables	Unadjusted Analysis		Adjusted Analysis	
	Odds Ratio (95% CI)	*p*-Value	Odds Ratio (95% CI)	*p*-Value
Age (for 1 year increase)	1.24 (0.6, 2.57)	0.560	1.08 (1.04, 1.12)	**<0.001**
Sex (female)	1.04 (1.00, 1.08)	**0.045**	1.29 (0.58, 2.89)	0.537
ASA <III vs. ≥III	3.19 (1.57, 6.47)	**0.001**	1.30 (0.62, 2.72)	0.482
Emergency	0.61 (0.29, 1.27)	0.185	—	—
Albumin	0.40 (0.22, 0.74)	**0.004**	0.44 (0.23, 0.85)	**0.014**
Hemoglobin	0.79 (0.66, 0.92)	**0.003**	0.93 (0.82, 1.07)	0.300
Baseline SBP	1.02 (1.00, 1.04)	0.057	—	—
IOH	3.00 (1.38, 6.51)	**0.005**	1.42 (0.54, 3.74)	0.478
PMA normalized by BSA (Reference: 5th quintile)	1.44 (1.19, 1.76)	**<0.001**	1.62 (1.37, 1.91)	**<0.001**
mFI score (categorical)	2.38 (1.12, 5.07)	**0.025**	0.95 (0.76, 1.19)	0.658
DM	1.31 (0.78, 2.19)	0.302	—	—
CHF	1.25 (0.72, 2.17)	0.433	—	—
HTN	1.57 (0.87, 2.82)	0.134	—	—
TIA or CVA	2.60 (1.51, 4.47)	**0.001**	—	—
Dependent functional status	1.29 (0.75, 2.22)	0.359	—	—
MI	1.85 (0.64, 5.49)	0.267	—	—
Peripheral disease	0.93 (0.42, 2.04)	0.853	—	—
CVA with sequelae	1.78 (0.92, 3.45)	0.088	—	—
Pulmonary disease	1.18 (0.64, 2.18)	0.591	—	—
Cardiac intervention or angina	1.36 (0.68, 2.73)	0.390	—	—
Impaired sensory	1.20 (0.54, 2.70)	0.655	—	—

CI, confidence interval; ASA, American Society of Anesthesiologists; SBP, systolic blood pressure; IOH, intraoperative hypotension; PMA, psoas muscle area; BSA, body surface area; mFI, modified frailty index; DM, diabetes mellitus; CHF, congestive heart failure; HTN, hypertension; TIA, transient ischemic attack; CVA, cerebrovascular disease; MI, myocardial infarction. *p*-values < 0.05 are in bold.

## Data Availability

The full trial protocol and the data supporting the findings of this study are available from the corresponding author upon request.

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
