# Peer review of "Predictability of Radiologically Measured Psoas Muscle Area for Intraoperative Hypotension in Older Adult Patients Undergoing Femur Fracture Surgery"

_jcm, 2023, doi:10.3390/jcm12041691_

Round 1

Reviewer 1 Report

The authors investigate whether there is an association of Psoas muscle cross sectional area and incidences of intraoperative hypotension in patients with hip fracture. The topic is of major interest, however, there are some issues which have to be addressed:

Methods:

·        Line 113: was the arterial line placed in awake patients?

·        Line 114: Which time interval is correct? Only a 1 minute interval is reasonable for such calculations

·        The definition of IOH is the major problem of the study. It is well known from the literature, that not only the incidences but also the duration an deepness of hypotension plays a major role in the development of adverse events. The authors therefore should take this considerations into account and calculate the Time weighted average below 65 mmHg for each patients to better compare the groups.

·        How was baseline blood pressure defined? The first value in the OR? Or much better, the value measured one day before on the evaluation?

Statistics:

·        What about not normally distributed variables? How was this tested?

Results:

·        In line 187, The authors describe median. However, a standard deviation is presented. This is not correct in case of not normally distributed variables. Here, the IQR should be presented.

·        Line 192. It is not surprising to see higher incidences of IOH in hypertensive patients give the definition of IOH as >30% drop in blood pressure. Therefore the TWA should be used for comparisons.

·        Table 1 shows too many patient confounders associated with IOH. Therefore, it is unlikely to clearly see a real association between frailty an IOH. The authors are strongly advised to match their patients in terms of age, hypertension, emergency, baseline blood pressure …. for investigating a relationship between frailty parameters and IOH. These confounders are therefore the reason for increased ICU admission rate, in other words: the patients are sicker, therefore develop more IOH. But we cannot say whether these patients need the ICU because of IOH.

Discussion:

·        The conclusion is not clear for me: We all know, that IOH is often a surrogate for sick patients, and many co-morbidities are associated with a worse outcome. Since the authors measured IOH inadequately (see above), the conclusion is not very valid. Moreover, we do not want to predict IOH but adverse outcome. This would be the right hypothesis. Comparison of adverse events and PMA.

Author Response

<Author response to reviewer 1>

We appreciate your time and valuable feedback on my manuscript. We are grateful to the reviewers for their insightful comments on my paper. Here is a point-by-point response to your valuable comments and concerns.

Methods:

# 1. Line 113: was the arterial line placed in awake patients?

Þ Response: The arterial line was placed under sedation before anesthesia or right after induction. Baseline blood pressure was measured by non-invasive blood pressure monitoring after they entered the operating room in a supine position, then induction was started. We monitored both invasive and non-invasive blood pressure during the surgery. I have revised the sentence as follows.

Þ Method, line 115: “The baseline blood pressure was defined as the first value of non-invasive blood pressure (systolic, diastolic, and mean blood pressure) after entering the operating room in the supine position. Radial artery cannulation was conducted under sedation before or right after induction, which was electronically recorded every five minutes (up to 1 min intervals in case of describing in detail) using BESTcare 2.0 (ezCaretech, Seoul, Korea) program. In some cases that attending anesthesiologists wanted to describe the hypotensive blood pressure in detail, it was recorded up to 1 min intervals.”

# 2. Line 114: Which time interval is correct? Only a 1-minute interval is reasonable for such calculations

Þ Response: I appreciated your thoughtful consideration. The majority of the time interval between vital signs was recorded at 5 min intervals regularly. In some cases that attending anesthesiologists wanted to describe the hypotensive blood pressure in detail, it was recorded up to 1 min intervals. To get time-weighted average data, we need to know an exact time period that the patient had hypotension. However, the retrospective design of the study limits the ability to get an exact hypotensive time period.

# 3. The definition of IOH is the major problem of the study. It is well known from the literature, that not only the incidences but also the duration and deepness of hypotension plays a major role in the development of adverse events. The authors therefore should take these considerations into account and calculate the Time-weighted average below 65 mmHg for each patient to better compare the groups.

Þ Response: As you were concerned, we wholeheartedly agree that the duration and the deepness of hypotension are important and play a major role in the development of adverse events. When we designed this study at first, we deliberated to calculate the time-weighted average. However, we couldn’t ensure how long hypotension lasted (up to 1-minute unit) for all patients because it was basically recorded every 5 minutes. This is an inherent limitation of the retrospective design, and several retrospective studies used dichotomous definitions of IOH which could not use the time-weighted average concept. [1-3]

 Based on previous studies, IOH durations ranging from 1 to 30 min increases the risk of myocardial and kidney injury [4,5,6] and mortality [7,8]. It was also reported that even a short duration of IOH was associated with acute kidney injury and myocardial injury [5]. A systemic review analyzed by Wesselink et al. [9] demonstrated that highly elevated risks of end-organ injury were reported for any exposure < 55-50 mmHg. In this context, we believe that our results which were not able to take the time factors into account can still provide valuable information. I hope you understand these limitations.

# 4. How was baseline blood pressure defined? The first value in the OR? Or much better, the value measured one day before on the evaluation?

Þ Response: We appreciate your meticulous comments. Baseline blood pressure was defined as the value measured in the OR just before giving anesthesia induction agents using non-invasive blood pressure monitoring. If the patients showed 30% higher baseline blood pressure than the average blood pressure during the hospitalization period, we allowed the blood pressure to be lowered after premedication with sedatives according to our institutions’ routine. In this case, blood pressure before the induction was considered baseline blood pressure. According to your advice, we added the definition of baseline blood pressure in Materials and Methods.

⇒ Method, line 121 “If the patients showed 30% higher baseline blood pressure than the average blood pressure during the hospitalization period, we allowed the blood pressure to be lowered after premedication with sedatives according to our institutions’ routine. In this case, blood pressure before the induction was considered baseline blood pressure.”

# 5. What about not normally distributed variables? How was this tested?

Þ Response: The distribution of data was analyzed to ensure that they conformed to the assumption of normality using Shapiro-Wilk tests. For variables that violated the assumption of normality (P < 0.05) such as input and estimated blood loss owing, we used Mann-Whitney U test. We have revised the sentence as follows.

Þ Statistics, line 169: “Continuous variables were analyzed using an independent t-test or Mann-Whitney U test after assessment for normality using Shapiro-Wilk test and are presented as mean ± standard deviation or median (interquartile range), as appropriate.” 

Results

# 6. line 187: The authors describe median. However, a standard deviation is presented. This is not correct in case of not normally distributed variables. Here, the IQR should be presented.

Þ Response: We accepted our error. Since it was a normally distributed variable, we corrected it to the mean value with standard deviation.

Þ Results, line 200, “The patients had a mean age of 81.34 ± 8.60 years, and 64% (n = 258) of the patients had ASA ≥ III.”

# 7. Line 192: It is not surprising to see higher incidences of IOH in hypertensive patients give the definition of IOH as >30% drop in blood pressure. Therefore, the TWA should be used for comparisons.

Þ Response: We fully understand your concern. Since our study population had a large portion of hypertensive patients, we considered that it is more desirable to apply both relative and absolute thresholds rather than just applying only the absolute threshold for patients with high blood pressure according to previous studies [4,10]. We also considered it would be ideal to measure the duration of hypotension together, however, as we mentioned above, there were limitations in the accuracy of measuring TWA_IOH using our institutions’ medical records in a retrospective-designed study.

# 8. Table 1 shows too many patient confounders associated with IOH. Therefore, it is unlikely to clearly see a real association between frailty and IOH. The authors are strongly advised to match their patients in terms of age, hypertension, emergency, baseline blood pressure …. for investigating a relationship between frailty parameters and IOH. These confounders are therefore the reason for increased ICU admission rate, in other words: the patients are sicker, therefore develop more IOH. But we cannot say whether these patients need the ICU because of IOH.

Þ Response: We agree with you that it is important to control confounding factors to access a causal relationship between frailty parameters (mFI score, PMA normalized by BSA) and IOH. As you recommended, it is ideal to perform propensity score matching (PSM) for variables that showed a great difference depending on whether or not IOH. However, in this study, it is not feasible to perform PSM because a large number of patients would be excluded from the analysis.

We consulted with a professional statistician and decided to perform inversed probability of treatment weighting (IPTW) to adjust for confounding factors [11]. We conducted IPTW to balance baseline patient characteristics in the exposed and unexposed groups by weighing each individual. The application of these weights to the study population creates a pseudopopulation in which measured confounders are equally distributed across groups. For IPTW covariates including sex, age, emergency op, chronic hypertension, and initial SBP, which showed a significant difference between the groups (Table 1.) were used for calculating propensity scores.

Frailty parameters

beta coefficient of IOH (β)

Standard error

P value

mFI score

0.077

0.126

0.032

PMA normalized by BSA (male)

0.639

0.091

<0.001

PMA normalized by BSA (female)

0.474

0.181

<0.001

We performed a linear regression to evaluate the relationships between frailty parameters (independent variables) and IOH (dependent variable) weighted by estimated propensity scores. The results indicated the relationship between IOH and frailty parameters (mFI score, PMA normalized by BSA) after adjusting the covariates level to make the groups similar by assigning individual weights. After being weighted by estimated propensity scores, the association between frailty parameters and IOH was still significant as presented in the table above. (P =0.032, < 0.001, < 0.001, respectively).

According to your advice, we added the results after conducting IPTW in Materials and Methods and Results.

Þ Method, line 184: “Confounding factors, including age, sex, albumin level, and ASA classification (< III or ≥ III), were identified from previous studies [28]. Additionally, significant variables, including 11 variables in mFI, were assessed by comparing patients with and without IOH. We also conducted inversed probability of treatment weighting (IPTW) to adjust for confounding factors [29]. Covariates which showed a significant difference in Table 1. were used for calculating propensity scores.”

We evaluated the incidence of ICU admission, hospital length of stay, and 3-month outcomes, after being weighted by estimated propensity scores. The results are presented in Table 2. and in the manuscript.

Table 2. Patients’ frailty assessment and postoperative 3-month outcomes before and after IPTW

ÞResults, line 239: “Patients in the IOH group showed a higher rate of ICU admission (35%; n = 100) than those in the no-IOH group (14.5%; n = 17; P < 0.001). After being weighted by estimated propensity scores, 125 patients (33.2%) in IOH group and 57 Patients (14.5%) in no-IOH group showed a rate of ICU admission (P < 0.001). Longer duration of hospital stay was higher in the IOH group (15.02 ± 10.00 days) than that in the no-IOH group (11.96 ± 6.08 days; P = 0.006). After being weighted by estimated propensity scores, the duration of hospital stay was significantly longer in the IOH group (14.60 ± 11.63 vs 12.46 ± 6.36 days; P = 0.002). The rates of 3-month adverse outcomes were significantly higher by 35.3 % (n = 101) in the IOH group than that in the no-IOH group by 10.2% (n = 12) (P < 0.001). After being weighted by estimated propensity scores, the rates of 3-month adverse outcomes were 37.1 % (n = 140) in the IOH group and 15.6 % (n = 62) in the no-IOH group (P < 0.001).”

Discussion:

# 9. The conclusion is not clear for me: We all know, that IOH is often a surrogate for sick patients, and many co-morbidities are associated with a worse outcome. Since the authors measured IOH inadequately (see above), the conclusion is not very valid. Moreover, we do not want to predict IOH but adverse outcomes. This would be the right hypothesis. Comparison of adverse events and PMA.

Þ Response: We appreciate your comment on this matter. While we acknowledge your concern, we wanted to note that the primary objective of this article is to identify the relationship between PMA and the occurrence of IOH. The most important matter for anesthesiologists during the surgery is dealing with IOH, although this is eventually a part of efforts to reduce postoperative adverse outcomes. Numerous studies (e.g., using risk factors, technology like hypotension prediction index (HPI), echocardiography of sonographic values, etc.) are still being conducted to get a better understanding of prediction in IOH and reduce its occurrence. In this context, we investigated whether PMA from CT images before surgery might be used as a simple preoperative method to assess the risk of IOH. Although we provided the relationship between PMA and postoperative 3-month adverse outcomes as secondary endpoints, it is hard to conclude that IOH is related to postoperative 3-month adverse outcomes from our results as shown in Table 4. It showed that PMA was a significant predictor of postoperative 3-month adverse outcomes (OR 1.62; 95% CI, 1.37–1.91; P <0.001), whereas presence of IOH was not a significant independent predictor of 3-month unfavorable outcomes after adjusting for confounders. (P = 0.478, Table 4). Further prospective studies considering the TWA_IOH are required to clarify the association between IOH and adverse surgical outcomes.

Therefore, in this study, we focused on IOH and investigated whether an easily measurable method, PMA, could predict IOH in older adult patients undergoing hip fracture surgery. We believe PMA is a valuable and practical method for predicting IOH and postoperative adverse outcomes, particularly in emergency clinical circumstances with restricted time and resources. I have revised the manuscript in Results and Discussion.

Þ To clarify our results and improve the validity of the conclusion, I revised and added the sentences in Introduction, line 58, Results, line 287, revised the sentence in Discussion, line 413, and the Conclusion, line 433.

Þ Introduction, line 58: “IOH is associated with postoperative outcomes and frequently occurs in frail patients; they typically have a higher sympathetic drive and reduced baroreflex sensitivity, which leads to IOH [20-21]. The association between low PMA and adverse surgical outcomes in older adult patients with hip fractures has been described [5]. To the best of our knowledge, no study has demonstrated the association between sarcopenia, presented as low PMA, and IOH development.”

Þ Results, line 287: “In addition, old age (OR 1.08; 95% CI, 1.04–1.12; P < 0.001) and low albumin levels (OR 0.44; 95% CI, 0.23–0.85; P = 0.014) were significantly associated with 3-month unfavorable outcomes, whereas presence of IOH was not a significant independent predictor of 3-month unfavorable outcomes after adjusting for confounders (P = 0.478).”

Þ Discussion, line 413: “As longer exposure to IOH may be related to an increased risk of postoperative outcomes, future prospective studies should be done to clarify the association between IOH and 3-month adverse outcomes considering the time-weighted average IOH.”

Þ Conclusion, line 433: “PMA normalized by BSA using pre-existing CT was a significant predictor of IOH occurrence and unfavorable surgical outcomes in older adult patients with hip fractures. Moreover, preoperative simple measurement of PMA was superior to mFI in predicting IOH. To discuss the relationship between IOH and adverse surgical outcomes, further prospective studies using the time weighted average IOH are warranted.”

References

[1] Sabate, S.; Mases, A.; Guilera, N.; et al. Incidence and predictors of major perioperative adverse cardiac and cerebrovascular events in non-cardiac surgery. Br J Anaesth 2011, 107, 879e90

[2] Jiang, X.; Chen, D.; Lou, Y.; Li, Z.; Risk factors for postoperative delirium after spine surgery in middle- and old-aged patients. Aging Clin Exp Res 2017, 29, 1039e44

[3] Vasivej, T.; Sathirapanya, P.; Kongkamol, C.; Incidence and risk factors of perioperative stroke in noncardiac, and nonaortic and its major branches surgery. J Stroke Cerebrovasc Dis 2016, 25, 1172e6

[4] Salmasi, V.; Maheshwari, K.; Yang, D.; et al. Relationship between intraoperative hypotension, defined by either reduction from baseline or absolute thresholds, and acute kidney and myocardial injury after noncardiac surgery: a retrospective cohort analysis. Anesthesiology 2017, 126 (1), 47–65.

[5] Micheal, W.; Philip, J. D.; Amit, X. G.; et al. Relationship between Intraoperative Mean Arterial Pressure and Clinical Outcomes after Noncardiac Surgery: Toward an Empirical Definition of Hypotension. Anesthesiology 2013, 119 (1), 507-515.

[6] Lukas, M. L.; Kaspar, F. B.; Marc, A. F.; et al. Impact of intraoperative hypotension on early postoperative acute kidney injury in cystectomy patients – A retrospective cohort analysis. J. Clin. Anesth. 2020, 66, 109906.

[7] Monk, T. G.; Saini, V.; Weldon, B. C.; et al. Anesthetic management and one-year mortality after noncardiac surgery. Anesth Analg 2005, 100 (1), 4-10.

[8] Bijker, J. B.; van Klei, W.A.; Vergouwe, Y.; et al. Intraoperative hypotension and 1-year mortality after noncardiac surgery.Anesthesiology 2009; 111 (1), 1217e26.

[9] Wesselink, E. M.; Kappen, T. H.; Torn, H. M.; et al. Intraoperative hypotension and the risk of postoperative adverse outcomes: a systematic review. Br J Anaesth 2018, 121(4),706-721.

[10] Choi, M. H.; Chae, J. S.; Lee, H. J.; et al. Pre-anaesthesia ultrasonography of the subclavian/infraclavicular axillary vein for predicting hypotension after inducing general anaesthesia: a prospective observational study. Eur J Anaesthesiol 2020, 37 (6), 474–481.

[11] Nicholas, C. C.; Viandsa, S. S.; Giovanini T.; et al. An introduction to inverse probability of treatment weighting in observational research. Clin 2022, 15(1), 14-20.

Reviewer 2 Report

The authors report the correlation of radiologically assessed psoas muscle area (PMA) and intraoperative hypotension (IOH) in elderly patients undergoing femur fracture surgery. The data presented are sound, and well worked up and the findings are interesting in various aspects. In the best-case scenario the data and conclusions drawn from the manuscript might lead to a somewhat personalized approach to preoperative patient preparation as well as personalized induction and fluid management of anesthesia for this kind of population and procedure. This might lead to an improved postoperative outcome if interpreted well.

However, my major concern with this paper is that the conclusions from the data analysis are not presented in an appropriate and clear form to guide the reader to clinically relevant conclusions. The authors themself stated in their results that "Multivariate logistic regression revealed that low PMA (OR 3.86; 95% CI, 2.82–5.29; P < 0.001) normalized by BSA, 250 high baseline systolic blood pressure (OR 1.03; 95% CI, 1.01–1.05; P = 0.001), and old age 251 (OR 1.06; 95% CI, 1.02–1.10; P = 0.003) were significant independent predictors of IOH, whereas the number of mFI variables was not a significant independent predictor of IOH 253 (P = 0.870)." So why do we need PSA then if age itself and preoperative high baseline blood pressure are proper predictors for IOH? In clinical routine age and baseline blood pressure can be assessed easily whereas PSA normalized by BSA is by far a more time-consuming procedure.

 Nevertheless, the data are extremely interesting. A conclusion that can really be drawn from the data is that PMA is superior in predicting IOH to mFI (Results: Figure 3). IOH has to be avoided since it is associated with negative perioperative outcomes. Could we avoid it with a different choice of drugs for induction of anesthesia and Hb and fluid management?  Induction of anesthesia was standardized and propofol is known to have negative inotropic effects besides vasodilatation. Could we use other drugs for patients at risk to develop IOH? The authors state that the transfusion trigger was a Hb of 8 mg/dL. Should the transfusion trigger be higher in patients with low PMA normalized by BSA? Could this lead to reduced IOH? Should patients with preoperatively low PMA receive special care after surgery?

Therefore my suggestions are:

It is far-fetched to associate IOH with 3 months outcome because many other factors influence outcomes during this prolonged postoperative period. IOH can reasonably be associated with neurological and cardiological complications within the immediate postoperative period (7 days). Please change and clarify throughout.

1. The title should be changed to "Predictability of Radiologically Measured Psoas Muscle Area for Intraoperative Hypotension and Adverse Outcome in Older Adult Patients Undergoing Femur Fracture Surgery".

2. Introduction: Why use 1-year mortality (line 35) in the introduction? Thereafter you refer to 3-month mortality which makes more sense. Please introduce a reference for 3-month mortality in the introduction.

3. Results: Is PMA normalized by BSA time-consuming to measure reliably? How long does it take to measure it? Can it be used easily in clinical routine? This would encourage clinicians to introduce PMA normalized by BSA in clinical routine, and thus, strengthen your manuscript. Please give a time estimate of the procedure in the results.

Did patients with IOH suffer increased perioperative neurological complications (TIA, stroke, postoperative delirium)? This would be very interesting if you can assess these data. If there is no difference IOH cannot be a major factor for the 3-months outcome.

Otherwise, the results are presented perfectly!

4. In the Discussion, it has clearly to be stated that:

- mFI is inferior to PMA normalized to BSA in detecting patients with perioperative high risk to develop IOH in elderly patients undergoing surgery for femur fracture. Therefore, it might be a valuable parameter to draw attention to these patients who are at an increased risk and act accordingly (intraoperative monitoring, induction of anesthesia, fluid management, and transfusion trigger).

- Sudfeld et al. found 34% of IOH, whereas your population revealed about 71% of IOH. This might be due to the definition of IOH or to the fact that Sudfeld et al. evaluated just 30 min after induction of anesthesia and your data refer to the entire period of the procedure. However, induction of anesthesia and the period till skin incision can be used for the optimization of hemodynamics since this period is not influenced by surgical interventions. Please discuss this accordingly.

Overall the manuscript is well-written and the data analysis is sound. The findings are very interesting to various physicians treating this patient population. 

Author Response

<Author response to reviewer 2>

# The authors report the correlation of radiologically assessed psoas muscle area (PMA) and intraoperative hypotension (IOH) in elderly patients undergoing femur fracture surgery. The data presented are sound, and well worked up and the findings are interesting in various aspects. In the best-case scenario the data and conclusions drawn from the manuscript might lead to a somewhat personalized approach to preoperative patient preparation as well as personalized induction and fluid management of anesthesia for this kind of population and procedure. This might lead to an improved postoperative outcome if interpreted well.

However, my major concern with this paper is that the conclusions from the data analysis are not presented in an appropriate and clear form to guide the reader to clinically relevant conclusions. The authors themself stated in their results that "Multivariate logistic regression revealed that low PMA (OR 3.86; 95% CI, 2.82–5.29; P < 0.001) normalized by BSA, 250 high baseline systolic blood pressure (OR 1.03; 95% CI, 1.01–1.05; P = 0.001), and old age 251 (OR 1.06; 95% CI, 1.02–1.10; P = 0.003) were significant independent predictors of IOH, whereas the number of mFI variables was not a significant independent predictor of IOH 253 (P = 0.870)." So why do we need PMA then if age itself and preoperative high baseline blood pressure are proper predictors for IOH? In clinical routine age and baseline blood pressure can be assessed easily whereas PMA normalized by BSA is by far a more time-consuming procedure.

 Nevertheless, the data are extremely interesting. A conclusion that can really be drawn from the data is that PMA is superior in predicting IOH to mFI (Results: Figure 3). IOH has to be avoided since it is associated with negative perioperative outcomes. Could we avoid it with a different choice of drugs for induction of anesthesia and Hb and fluid management?  Induction of anesthesia was standardized and propofol is known to have negative inotropic effects besides vasodilatation. Could we use other drugs for patients at risk to develop IOH? The authors state that the transfusion trigger was a Hb of 8 mg/dL. Should the transfusion trigger be higher in patients with low PMA normalized by BSA? Could this lead to reduced IOH? Should patients with preoperatively low PMA receive special care after surgery?

Þ Response: We appreciated your thoughtful consideration. As you said, some patients’ characteristics such as age and preoperative blood pressure have been shown to be associated with IOH. However, it does not often give enough clue to predict IOH (For example, from our results, age was a significant factor to predict IOH, however, the mean age of no-IOH group and IOH was 78.62, and 82.46, all of which represent old patients). Hence, lots of studies (e.g., using risk factors, technology like HPI, echocardiography of sonographic values, etc.) have been still conducted to get a better sense of prediction in intraoperative or postoperative outcomes.

In our institution, we have measured frailty using questionnaire-based methods before some major surgeries, while frailty has been spotlighted again for its usefulness as a preoperative risk evaluation tool in the recent decade. It is actually time-consuming for clinical use and limited to applicate to patients with femur fractures owing to their restricted physical conditions or inability to provide medical history as we mentioned in the text. However, PMA measurement obtained from existing CT images can be is a feasible method like anesthesiologists checking chest x-rays before surgery. More interestingly, our data showed that PMA could predict IOH better than mFI score. We believe that simple measurement of PMA from CT images will help evaluate and understand patients preoperatively. I really appreciate your thoughtful comments.

Therefore, my suggestions are:

# It is far-fetched to associate IOH with 3 months outcome because many other factors influence outcomes during this prolonged postoperative period. IOH can reasonably be associated with neurological and cardiological complications within the immediate postoperative period (7 days). Please change and clarify throughout.

Þ Response: I agree with you that there might be various factors that can influence outcomes during this prolonged postoperative period. We have added references for using 3-month outcome data according to your valuable comment #2. Although we provided postoperative adverse outcomes as secondary endpoints, it is hard to conclude the relationship between IOH and postoperative outcomes from our results. Since our primary outcome was to investigate the association with IOH, it would be underpowered to conclude the association with postoperative outcomes which has much lower incidence compared to IOH.

#1. The title should be changed to"Predictability of Radiologically Measured Psoas Muscle Area for Intraoperative Hypotension and Adverse Outcome in Older Adult Patients Undergoing Femur Fracture Surgery".

Þ Response: We appreciated your thoughtful consideration. However, this article provides the causational relationship between the PMA and the likelihood of IOH occurrence. Although we provided the relationship between PMA and postoperative 3-month adverse outcomes as secondary endpoints, it is hard to conclude that IOH is related to postoperative 3-month adverse outcomes from our results. Because the result in Table 4. showed that IOH was not a significant independent predictor of 3-month unfavorable outcomes after adjusting for confounders. (P = 0.478).

Therefore, in this study, we focused on IOH and investigated whether an easily measurable method, PMA, could predict IOH in older adult patients undergoing hip fracture surgery. We would like to revise the introduction and conclusion to improve validity rather than changing the title.

According to our hypothesis, we revised the sentences to clarify the introduction in line 58, and the conclusion in line 433.

Þ Introduction, line 58: “IOH is associated with postoperative outcomes and frequently occurs in frail patients; they typically have a higher sympathetic drive and reduced baroreflex sensitivity, which leads to IOH [20-21]. The association between low PMA and adverse surgical outcomes in older adult patients with hip fractures has been described [5]. To the best of our knowledge, no study has demonstrated the association between sarcopenia, presented as low PMA, and IOH development.”

Þ Conclusion, line 433: “PMA normalized by BSA using pre-existing CT was a significant predictor of IOH occurrence and unfavorable surgical outcomes in older adult patients with hip fractures. Moreover, preoperative simple measurement of PMA was superior to mFI in predicting IOH. To discuss the relationship between IOH and adverse surgical outcomes, further prospective studies using the time weighted average IOH are warranted.”

#2. Introduction: Why use 1-year mortality (line 35) in the introduction? Thereafter you refer to 3-month mortality which makes more sense. Please introduce a reference for 3-month mortality in the introduction.

Þ Response: As you recommended, I changed 1-yr mortality to 3-month mortality rate and introduced the reference [2-4] as below.

Þ Introduction, line 36: “Surgical outcomes of patients with hip fractures are associated with a high 3-month mortality rate (4.7–19.5 %) and physical morbidity [2-4].”

References were added as below.

  1. Ove, T.; Fredrik, H.; Ola, E, D.; Are,H, P.; Olav, R. Clinical and biochemical prediction of early fatal outcome following hip fracture in the elderly. Int Orthop 2011, 35, 903–907.
  2. Istianah, U.; Nurjannah, I.; Magetsari, R. Post-discharge complications in postoperative patients with hip fracture. J Clin Orthop Trauma 2021,14, 8–13.
  3. Ha, Y.; Cha Y.; Yoo, J.; Lee, J.; Lee Y.; Koo K. Effect of dementia on postoperative mortality in elderly patients with hip fracture. J Korean Med Sci 2021, 36, e238.

#3. Results: Is PMA normalized by BSA time-consuming to measure reliably? How long does it take to measure it? Can it be used easily in clinical routine? This would encourage clinicians to introduce PMA normalized by BSA in clinical routine, and thus, strengthen your manuscript. Please give a time estimate of the procedure in the results.

Þ Response: Thank you for your thoughtful consideration. As I mentioned in the discussion, measuring PMA from CT images took just a few minutes. PACS provides a function to measure an area like PMA using region of interest (ROI) program, and we just need to draw an outline of the psoas muscle as shown in Fig 2, which can be easily done within a minute. Then, calculating by BSA (√ [height (cm) * weight (kg)]/ 3600) can be also readily done with a calculator within a minute. For us, it took less than one minute to get the result for one patient. According to your advice, we have added the estimation time in the results.

Þ Results, Line 224: “The ICC between the two researchers for measuring PMA was excellent: 0.977 (95% CI, 0.95–0.99; P < 0.001), and measuring PMA from CT images took only less than 1min.”

#4. Did patients with IOH suffer increased perioperative neurological complications (TIA, stroke, postoperative delirium)? This would be very interesting if you can assess these data. If there is no difference IOH cannot be a major factor for the 3-months outcome.

Þ Response: As we described postoperative data in Table 2, postoperative neurologic complication (CVA) showed higher incidence in IOH group, but there was no significant difference between the groups. Also, CVA was not the significant factor after adjusting confounding factors (Table 3). From the previous studies [1,2], the incidence of stroke was reported as 0.004% to 0.09% within 10 days. They analyzed 48,241 [1] and 3387 patients [2] for assessing the relationship between IOH and stroke. Our study did not have enough sample size to compare and conclude the association with neurologic complications, because we need a much greater sample size to compare postoperative outcomes that have much lower incidence compared to that of IOH. Therefore, no firm conclusion on the relationship between IOH exposure and neurologic complications or 3-month adverse outcomes can be drawn from this study. To conclude the relationship between IOH exposure and neurologic complications, further studies with a greater number of patients are warranted.

#5. In the Discussion, it has clearly to be stated that: mFI is inferior to PMA normalized to BSA in detecting patients with perioperative high risk to develop IOH in elderly patients undergoing surgery for femur fracture. Therefore, it might be a valuable parameter to draw attention to these patients who are at an increased risk and act accordingly (intraoperative monitoring, induction of anesthesia, fluid management, and transfusion trigger).

Þ Response: Thank you for your thoughtful consideration. We tried to demonstrate the usefulness of PMA instead of mFI, but it seems that my intentions were not conveyed properly. Therefore, I revised the sentences according to your advice.

ÞIn Line 340: “In the present study, we found that PMA normalized by BSA was superior to mFI in predicting IOH. Measuring PMA from CT images before surgery could be a faster and more simple method than mFI for assessing the risks of IOH. Predicting IOH could improve individual anesthetic management, in terms of preparing for hypotensive situations: meticulous titration of anesthetic agents, preparation of additional peripheral intravenous lines, vasopressors, sufficient pre-induction fluid supplement, and continuous arterial blood pressure monitoring to prevent hypotensive events during the surgery.”

#6. Sudfeld et al. found 34% of IOH, whereas your population revealed about 71% of IOH. This might be due to the definition of IOH or to the fact that Sudfeld et al. evaluated just 30 min after induction of anesthesia and your data refer to the entire period of the procedure. However, induction of anesthesia and the period till skin incision can be used for the optimization of hemodynamics since this period is not influenced by surgical interventions. Please discuss this accordingly.

Þ Response: As you mentioned, IOH incidence differs by large variations in the definition of IOH and the characteristics of the study population. Nirav et al. [3] reported that 88% of the moderate to high-risk population had at least one hypotensive event as defined as MAP <65 mmHg for 1 min, which had a higher incidence than our results.

Even though the post-induction period is not influenced by surgical interventions, it can be also affected by other factors like patients’ dehydration status, the amount, or kinds of induction agents, and whether or not taking antihypertensive medication. We considered that hypotensive events in the entire surgical process are more clinically important to our population because we believed that they reflected perioperative patients’ overall condition more than post-induction hypotension (46.7% of our population, not presented in the manuscript).

# Overall, the manuscript is well-written and the data analysis is sound. The findings are very interesting to various physicians treating this patient population.

Þ Response: Thank you for your kind comments. I hope the revised manuscript would be acceptable. I appreciate your consideration.

References

[1] Bijker, J.B.; Persoon, S.; Peelen, L.; et al. Intraoperative hypotension and perioperative ischemic stroke after general surgery. Anesthesiology 2012; 116: 658e64.

[2] Sabate’, S.; Mases, A.; Guilera, N.; et al. Incidence and predictors of major perioperative adverse cardiac and cerebrovascular events in non-cardiac surgery. Br J Anaesth 2011, 107(1), 879e90.

[3] Nirav, J. S.; Graciela, M.; Sachin K. The incidence of intraoperative hypotension in moderate to high risk patients undergoing non-cardiac surgery: A retrospective multicenter observational analysis. J Clin Anesth 2020 Nov;66:109961

Round 2

Reviewer 1 Report

Concerns were adequately addressed.